# Difluorocarbene-induced [1,2]- and [2,3]-Stevens rearrangement of tertiary amines

Jianke Su[1,5], Yu Guo[1,5], Chengbo Li[1] & Qiuling Song [1,2,3,4] ✉

The [1,2]- and [2,3]-Stevens rearrangements are one of the most fascinating chemical bond reorganization strategies in organic chemistry, and they have been demonstrated in a wide range of applications, representing a fundamental reaction tactic for the synthesis of nitrogen compounds in chemical community. However, their applicabilies are limited by the scarcity of efficient, general, and straightforward methods for generating ammonium ylides. Herein, we report a general difluorocarbene-induced tertiary amine-involved [1,2]- and [2,3]-Stevens rearrangements stemmed from in situ generated difluoromethyl ammonium ylides, which allows for the rearrangements of versatile tertiary amines bearing either allyl, benzyl, or propargyl groups, resulting in the corresponding products in one reaction under the same reaction conditions with a general way. Broad substrate scope, simple operation, mild reaction conditions and late-stage modification of natural products highlight the advantages of this strategy, meanwhile, this general rearrangement reaction is believed to bring opportunities for the transformations of nitrogen ylides and the assembly of valuable tertiary amines and amino acids. This will further enrich the reaction repertoire of difluorocarbene species, facilitate the development of reactions involving difluoromethyl ammonium salts, and provide an avenue for the development of this type of rearrangement reactions.

Tertiary amines and their derivatives are widely present in natural products and bioactive molecules as an important functionality, and it is beneficial to achieve the precise late-stage modification of existing drugs and natural products, without significantly altering the parental scaffolds and maintaining the functionalities[1–5]. The Stevens rearrangement is typically applied to construct complex nitrogen/sulfur-containing compounds through [1,2]- or [2,3]-sigmatropic rearrangement, starting from ammonium or sulfonium salts. However, harsh conditions, such as the presence of a strong base, are usually necessary and unavoidable, and the key intermediate is nitrogen[6–14] or sulfur[15–19] ylide. Therefore, Stevens rearrangement is often considered a commonly used molecular editing method for prevalent tertiary amines. Usually, the methods of producing nitrogen ylides mainly include the following strategies: prepared from tertiary amines within situ-generated arynes[20–22] or alkyl halides, or activated by equivalent of Lewis acids[23–27] (Fig. 1A). For [1,2]-Stevens rearrangement, it typically involves the migration of benzyl substituents on tertiary amines to the adjacent carbon atom through an [1,2]-sigmatropic rearrangement mechanism[28,29], or the ring expansion reactions of small cyclic amines with diazo compounds, as reported by Lacour and Feng et al.[30–32]. For [2,3]-sigmatropic rearrangement, transition-metal-catalyzed methods were also developed for the generation of nitrogen ylides[33–40], for

[1]Institute of Next Generation Matter Transformation, College of Material Sciences Engineering, Huaqiao University, Xiamen, Fujian 361021, China. [2]Key Laboratory of Molecule Synthesis and Function Discovery, Fujian Province University, College of Chemistry at Fuzhou University, Fuzhou, Fujian 350108, China. [3]State Key Laboratory of Organometallic Chemistry, Shanghai Institute of Organic Chemistry, Chinese Academy of Sciences, Shanghai 200032, China. [4]School of Chemistry and Chemical Engineering, Henan Normal University, Xinxiang, Henan 453007, China. [5]These authors contributed equally: Jianke Su, Yu Guo. ✉e-mail: qsong@fzu.edu.cn

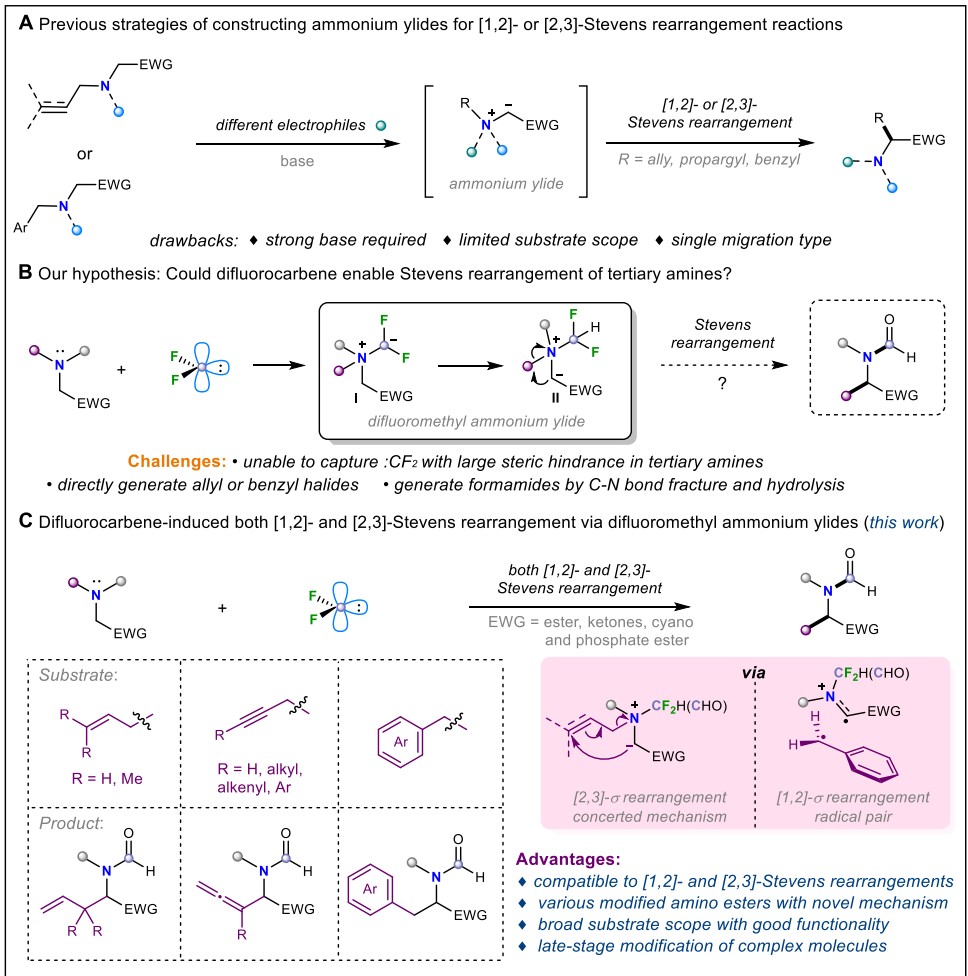

**Fig. 1 | Known strategies for Stevens rearrangements of tertiary amines and the progress of difluoromethyl ammonium salts. A** Previous strategies of constructing ammonium ylides for [1,2]- or [2,3]-Stevens rearrangement. **B** Our hypothesis: could difluorocarbene enable Stevens rearrangement of tertiary amines? **C** Difluorocarbene-induced both [1,2]- and [2,3]-Stevens rearrangement via difluoromethyl ammonium ylides (this work).

instance, a Pd-catalyzed allylic amination strategy using tertiary amino esters and allyl carbonates for the generation of nitrogen ylides has been disclosed by Tambar group in 2011. These reactive intermediates (arynes[20–22], π-Allyl palladium complex[36], etc.) enable the Stevens rearrangement to proceed under milder conditions, avoiding the harsh conditions such as high temperature and strong base required by the reaction previously. Tertiary allyl amines with diazoesters via a Rh-catalyzed cross-coupling for direct access to nitrogen ylides have emerged as a universal method. In 2021, Zhang and coworkers demonstrated a *5-endo-dig* cyclization of tertiary amines with intra-molecular alkynes activated by π-Lewis-acid to form quaternary ammonium salts, then render nitrogen ylides[41]. However, the uni-versalities of the known Stevens rearrangements are quite poor, usually it is difficult for them to be compatible with all allyl, propargyl, and benzyl groups in one reaction under the same conditions[6–14], which thus restricts the derivation of product diversity. Consequently, studies toward the efficient and versatile methods for both [1,2]- and [2,3]-Stevens rearrangements are highly desirable.

Given the narrow substrate scopes and limited migration types in the previous strategies, we envisioned discovering an efficient and powerful approach, which can start with simple starting materials to achieve both [1,2]-and [2,3]-Stevens rearrangements and be compa-tible with all of allyl, propargyl, and benzyl groups smoothly in the same reaction conditions. Based on the structure of ammonium ylides[10–15], we focus our attention on the difluoromethyl ammonium

salts, which is inspired by our previous research on the reactivity of difluorocarbene[42,43]. Since this structure was proposed by our group in 2020[44], it has been extensively studied, and this key intermediate can be used to achieve the construction of complex fluorinated compounds[45–48], the synthesis of natural products[49,50], and deamina-tive Suzuki cross-coupling reactions without transition-metal catalysis[51]. Based on its exceptional performance and enormous application potential, we envisioned whether [1,2]- and [2,3]-Stevens rearrangement reactions could be performed on the special in situ-generated difluoromethyl ammonium ylide **I** in the presence of base, since this ylide **I** could be readily converted to a difluoromethyl-bearing nitrogen ylide **II** with water[52,53]. Moreover, since the difluor-omethyl quaternary ammonium ylide **II**[54,55] also exhibits electron-withdrawing characteristics, this property may potentially facilitate Stevens rearrangement even more smoothly. However, there are some inherent challenges in this hypothesis: (a) When substituents with large steric hindrance are on tertiary amines, it might be difficult for nitro-gen atom to capture difluorocarbene species, resulting in the failure of the reaction. (b) When difluoromethyl ammonium salt contains allyl or benzyl groups, an $S_N2$ pathway might take place by the nucleophilic reagents (for instance, halogen ions) in the system, resulting in the formation of benzyl or allyl halides, thus delaying the occurrence of Stevens rearrangement. (c) Difluoromethyl ammonium salt might convert into a formylation product[56,57] in the presence of water, thereby preventing the rearrangement reaction from occurring

**Table 1 | The condition screening for the difluorocarbene-induced tertiary amine-involved Stevens rearrangement**

| Entries | Solvent | Base (X equiv.) | $[:CF_2]$ (3 equiv.) | Yield (%)[a] |
|---|---|---|---|---|
| 1 | $CH_3CN$ | $K_2CO_3$ (3) | $BrCF_2COOK$ (**2a**) | 45[b] |
| 2 | $CH_3CN$ | $K_2CO_3$ (3) | $BrCF_2COONa$ (**2b**) | 35 |
| 3 | $CH_3CN$ | $K_2CO_3$ (3) | $BrCF_2COOEt$ (**2c**) | 25 |
| 4 | $CH_3CN$ | $K_2CO_3$ (3) | $ClCF_2COONa$ (**2d**) | 51 |
| 5 | $CH_3CN$ | $K_2CO_3$ (3) | $ClCF_2H$ (**2e**) | trace |
| 6 | $CH_3CN$ | HCOONa (3) | $ClCF_2COONa$ (**2d**) | trace |
| 7 | $CH_3CN$ | $Rb_2CO_3$ (3) | $ClCF_2COONa$ (**2d**) | 64 |
| 8 | $CH_3CN$ | LiOH (3) | $ClCF_2COONa$ (**2d**) | trace |
| 9 | $CH_3CN$ | $Na_2CO_3$ (3) | $ClCF_2COONa$ (**2d**) | 31 |
| 10 | $CH_3CN$ | $Et_3N$ (3) | $ClCF_2COONa$ (**2d**) | ND |
| 11 | $CH_3CN$ | $K_3PO_4$ (3) | $ClCF_2COONa$ (**2d**) | 85[b] |
| 12 | $CH_3CN$ | $K_3PO_4$ (3) | **2d** (1 equiv.) | 71 |
| 13 | $CH_3CN$ | $K_3PO_4$ (3) | **2d** (1.5 equiv.) | 86 |
| 14 | $CH_3CN$ | $K_3PO_4$ (1.5) | **2d** (1.5 equiv.) | 55 |
| 15 | $CH_3CN$ | $K_3PO_4$ (2) | **2d** (1.5 equiv.) | 71 |
| 16 | $CH_3CN$ | $K_3PO_4$ (2.5) | **2d** (1.5 equiv.) | 78 |

Reaction condition: [a]**1** (1 equiv., 0.2 mmol), **2** (3 equiv., 0.6 mmol), base (3 equiv.), $CH_3CN$ (2 mL) at 90 °C for 12 h under argon; GC yields; [b]isolated yields; [c]*ND* not detected.

(Fig. 1B). To our delight, when we used difluorocarbene species to induce Stevens rearrangement, we could smoothly obtain the expected product, and it effectively avoids the occurrence of the aforementioned side reactions. Herein, we report an efficient difluorocarbene-induced Stevens rearrangement reaction. Mild reaction conditions, wide substrate range, diverse migration modes, multiple product types, and the extensive modification of various natural products all convincingly demonstrate the advantages of this method. This will further enrich the reaction repertoire of difluorocarbene species, facilitate the development of reactions involving difluoromethyl ammonium salts, and provide an avenue for the development of this type of rearrangement reaction. In such a difluorocarbene-induced rearrangement reaction, it can accommodate various types of rearrangements with different substrates, significantly expanding the utility and product diversity of this reaction. When the substituent on nitrogen atom is a benzyl group, [1,2]-rearrangement proceeds smoothly. When the substituent is an alkyne or an alkene, [2,3]-sigmatropic rearrangement can occur efficiently, and after alkyne rearrangement, 1,1-disubstituted allenes are readily obtained (Fig. 1C).

## Results
### Reaction optimization
To test the feasibility of the hypothesis, we set out to probe the reaction conditions using tertiary amine **1a**, $BrCF_2COOK$ (**2a**) as tertiary amine activating reagent, $K_2CO_3$ as the base in the presence of $CH_3CN$ (Table 1). The desired Stevens rearrangement product **3a** was obtained in 45% isolated yield. Encouraged by the above results,

we then proceeded to further optimize the reaction conditions, types, and dosages of difluorinated reagents, bases, and reaction temperatures on the product yields, were emphatically investigated in Table 1. We examined originally the effects of different difluorocarbene reagents on this reaction, the results indicated that among $BrCF_2COOK$ (**2a**), $BrCF_2COONa$ (**2b**), $BrCF_2COOEt$ (**2c**), $ClCF_2COONa$ (**2d**), $ClCF_2COONa$ (**2e**), $ClCF_2COONa$ (**2d**) demonstrated the best performance and the desired product **3a** was obtained in 51% yield (entries 1–5, Table 1). Subsequently, further investigations were conducted on the effects of different types of bases on the difluorocarbene-induced Stevens rearrangement, which suggested that $K_3PO_4$ exhibited the best overall balance of activity and basicity among HCOONa, $Rb_2CO_3$, LiOH, $Na_2CO_3$, $Et_3N$, as well as $K_3PO_4$ (entries 6–11, Table 1), the yield of **3a** increased to 85%. The dosage of difluorinated reagents and base were also assessed (entries 12–16, Table 1), a better outcome was obtained when 1.5 equiv of $ClCF_2COONa$ (**2d**) was employed instead of 3 equiv. Base evaluations suggested that 3 equiv. of $K_3PO_4$ is necessary to ensure the high yield of this rearrangement reaction (see Supplementary Information for details).

### Substrate scope
After identifying the optimal conditions, we turned to evaluate the substrate scope for this difluorocarbene-induced Stevens rearrangement reaction (Fig. 2). We investigated the effect of ester groups as electron-withdrawing groups on the rearrangement reaction. Different alcohol-derived esters including methyl, ethyl, *tert*-butyl and some

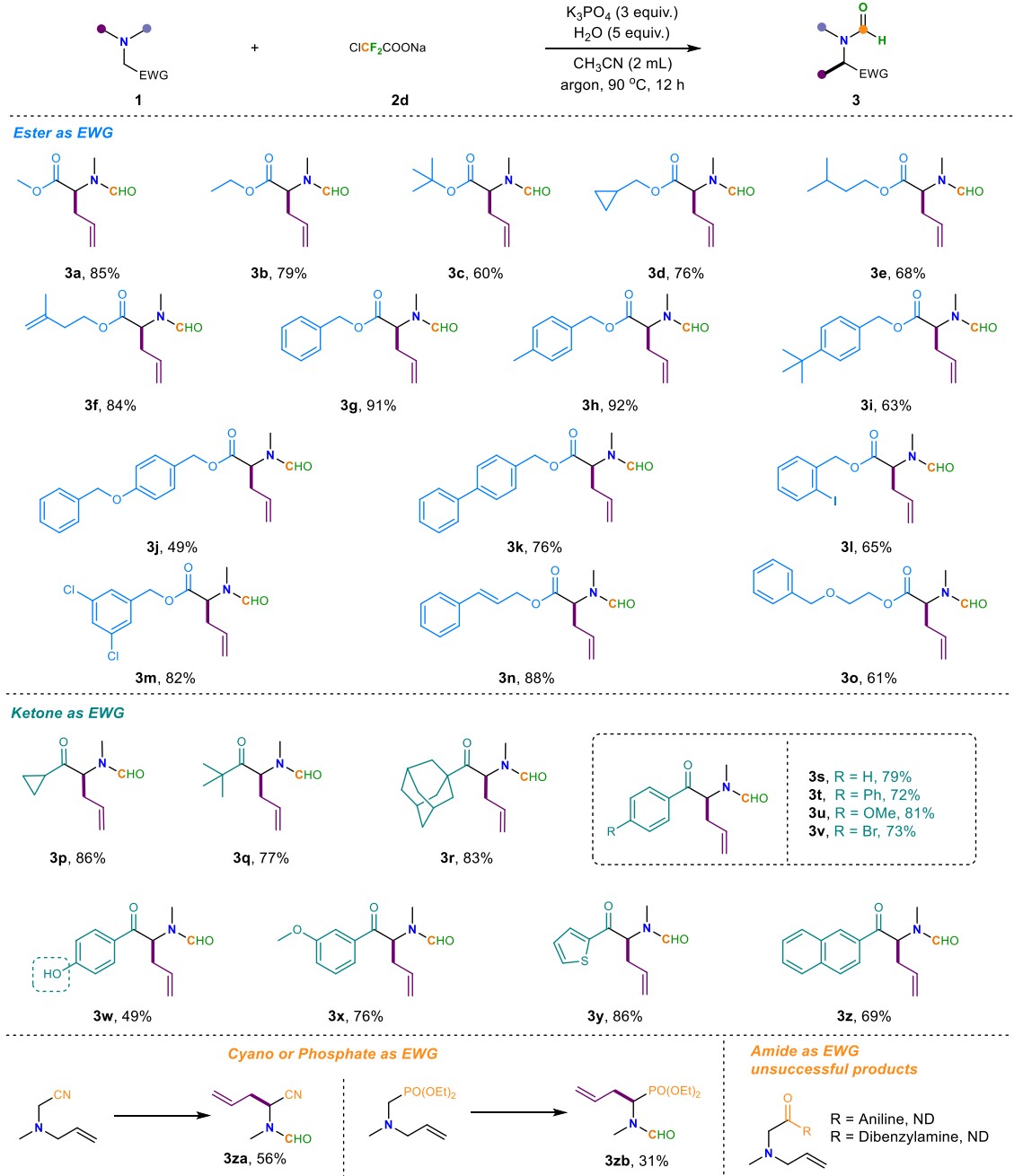

**Fig. 2 | Difluorocarbene-induced allylic Stevens rearrangement of tertiary amines.** Reaction condition: [a]**1** (1 equiv., 0.2 mmol), **2d** (1.5 equiv., 0.3 mmol), $H_2O$ (5 equiv.), $K_3PO_4$ (3 equiv.), $CH_3CN$ (2 mL) at 90 °C for 12 h under argon. ND not detected.

other alkyl groups were all well tolerated in this rearrangement reaction, and afforded the corresponding coupling products **3a–3f** in 68-85% yields, respectively. Different benzyl as the ester derivatives were also compatible substituents for this transformation (**1g-1o**), the corresponding product (**3g-3o**) was obtained in good to excellent yields. Notably, this protocol also featured an admirable scope with respect to ketone substrates. The corresponding target products **3p-3z** were all smoothly delivered (69–86% yields), no matter that it was an aliphatic ketone or an aryl ketone. Aliphatic ketones, such as cyclopropyl (**1p**), *tert*-butyl (**1q**) and adamantly (**1r**) were all delivered the corresponding rearrangement products (**3p-3r**) with a moderate to excellent yields. For the aromatic ones, substrates bearing electron-neutral (**1s-1t**), electron-rich (**1u**), as well as electron-deficient substituents (**1v**) at the *para*-position of the aromatic rings all furnished the desired

rearrangement products (**3s-3v**) smoothly. Of note, this reaction was even not greatly disturbed when a free hydroxyl group at the *para*-position of the aromatic rings, the final product **3w** still could be obtained in decent yield. In addition, the *meta*-methoxy substrate **1x** is also a good substrate for this transformation. This study was also easily extendable to the heteroaromatic ketone such as thiophene (**3y**). Fused ring reactant like 1-naphthaldehyde (**1z**) was also a suitable candidate for this transformation and the corresponding product **3z** was obtained in 69% yield. We also made attempts with other electron-withdrawing groups and found that both cyano and phosphate esters were compatible with our reaction system. They afforded the allylic rearranged products with yields of 56% and 31%, respectively. When employing an amide as the electron-withdrawing group, it failed to induce the Stevens rearrangement reaction. Instead, we observed the

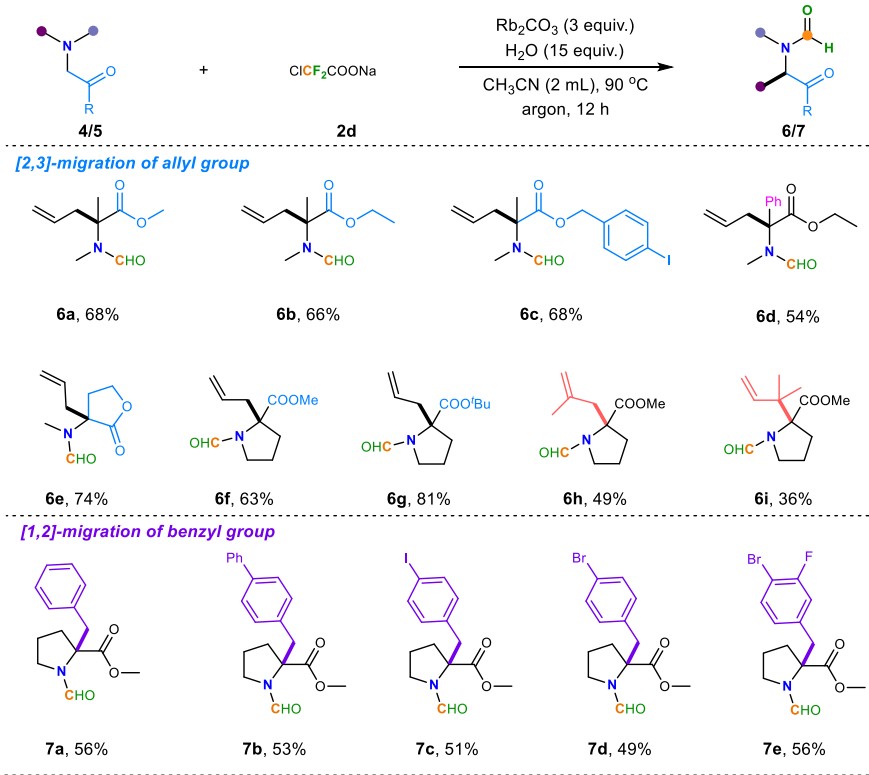

**Fig. 3 | Difluorocarbene-induced allylic and benzyl Stevens rearrangement of tertiary amines.** Reaction condition: [a]**4/5** (1 equiv., 0.2 mmol), **2d** (1.5 equiv., 0.3 mmol), H$_2$O (15 equiv.), Rb$_2$CO$_3$ (3 equiv.), CH$_3$CN (2 mL) at 90 °C for 12 h under argon.

formation of a product where the tertiary amine undergoes direct allylic cleavage at the nitrogen atom upon interaction with difluorocarbene, followed by formylation.

Encouraged by the above results, we further explored the tolerance range of amine substrates. Since many amino acid derivatives typically include a polysubstituted skeleton, we then wondered whether this Stevens rearrangement can also be employed for such poly-substituted amino esters. Not surprisingly, when these substrates were subjected to the previous standard conditions it was found that the corresponding target products could only be obtained in 25% yield. We replaced Rb$_2$CO$_3$ as the base and found that the amount of water had a significant impact on the reaction outcomes (See Supplementary Table 5 in Supplementary Information for details). After simple condition reevaluations, expectedly, these substrates were equipotential to afford the corresponding modified amino ester derivatives in moderate to excellent yields under the marginally reoptimized conditions (Fig. 3). For such polysubstituted amino esters, different esters, including both aliphatic and benzylic ones all smoothly achieved this transformation in decent yields (**6a-6d**). For cyclic amino esters, like lactone (**4e**) and derivatives of proline (**4f-4h**), it suggests that they were all suitable candidates for this transformation and the corresponding polysubstituted amino ester derivatives (**6e-6h**) were obtained smoothly. More importantly, when internal alkene **4i** was used as a substrate, it underwent the Stevens rearrangement to yield the terminal alkene product **6i**, which provides further evidence on the occurrence of a [2,3]-sigmatropic rearrangement process in this reaction. Of note and fortunately, benzyl group was also compatible with the difluorocarbene-induced rearrangement reaction, resulting in benzyl [1,2]-sigmatropic rearrangement of ammonium ylides. Next, *N*-benzyl with various substituents on the benzene ring attached to the amino esters was explored (**5a-5e**), various

functional groups were tolerable in the reaction, and delivered the corresponding products **7a-7e** in decent yields.

After examining the substrate scope of allyl and benzyl Stevens rearrangement reactions, we then set out to evaluate the generality of this protocol for propargyl tertiary amines (Fig. 4). It was found that esters bearing variations on the *N*-propargyl units were perfectly compatible with the reaction conditions, which further indicates that the reaction has a wide range of substrate tolerance. Notably, an *N*-propargyl substrate with a terminal acidic proton (i.e., R = H) also underwent a smooth conversion to the desired product **9a** in 82% yield. And for the *tert*-butyl substituted amino ester **8b**, it could also be able to deliver the corresponding products **9b** in a good yield. Meanwhile, the propargyl substrates, which bear different alkyl and aryl groups were also good candidates for the rearrangement. For instance, this reaction worked well with methyl, ethyl, cycloalkyl, amyl, and tert-butyl-substituted substrates (**8c-8g**), giving the corresponding products **9c-9g** in moderate to good yields. Alkyne derivatives substituted with pyran (**8 h**) and piperidine (**8i**) were also suitable substrates, which can also render the corresponding products **9h–9i** in the yields of 77% and 62%, respectively. For alkyne compounds containing unprotected alkyl alcohols (**8j**), the final product **9j** was successfully obtained without yield erosion, which also indicates that the reaction has good functional group tolerance. For 1, 3-enyne compounds, they were also good substrates for this rearrangement reaction, both terminal alkene **8k** and internal cyclic alkene **8l** could be smoothly converted into the target products **9k-9l** with good yields accordingly. *N*-Propargyl substrates bearing electron-neutral (-H), electron-donating (-Et, and -OMe), and electron-withdrawing (-Cl, -Br, and -CHO) groups at the *para* position of benzene rings all furnished the desired allene products (**9m-9r**) smoothly. Of note, the structure of compound **9p** was unambiguously confirmed by X-ray crystal analysis.

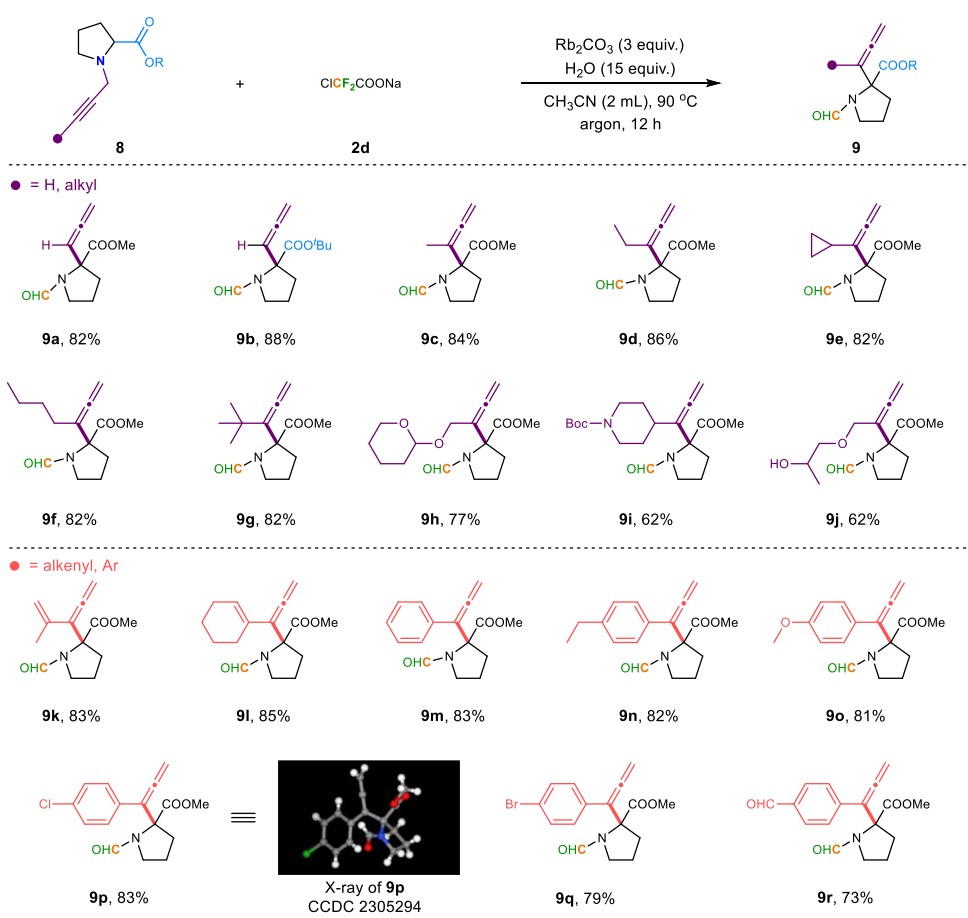

**Fig. 4 | Difluorocarbene-induced propargyl Stevens rearrangement of tertiary amines.** Reaction condition: [a]**8** (1 equiv., 0.2 mmol), **2d** (3 equiv., 0.6 mmol), H$_2$O (15 equiv.), Rb$_2$CO$_3$ (3 equiv.), CH$_3$CN (2 mL) at 90 °C for 12 h under argon.

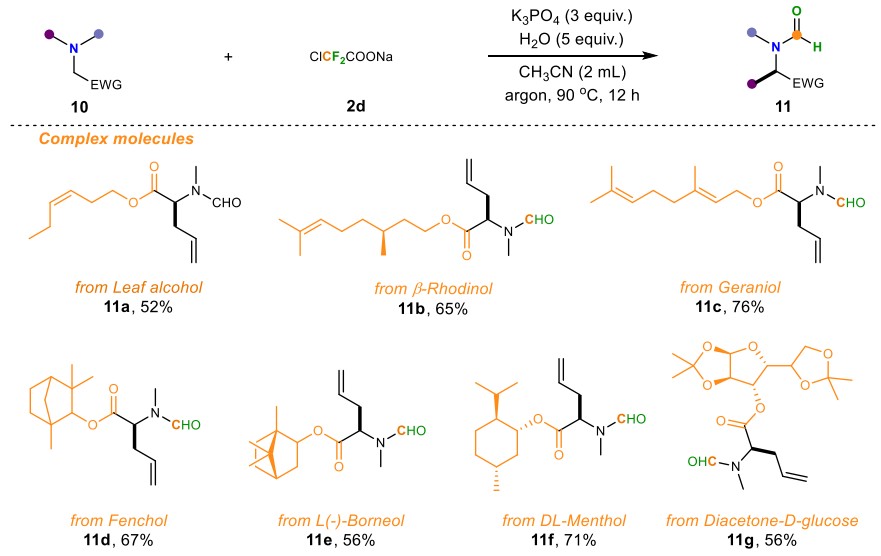

**Fig. 5 | Scope of substrates derived from complex molecules.** Reaction condition: [a]**10** (1 equiv., 0.2 mmol), **2d** (1.5 equiv., 0.3 mmol), H$_2$O (5 equiv.), K$_3$PO$_4$ (3 equiv.), CH$_3$CN (2 mL) at 90 °C for 12 h under argon.

Tertiary amines are also the core backbone of many natural product molecules. The above results clearly demonstrated that our method has a very broad substrate scope and wide functional group compatibility. Subsequently, we then focused on exploring the substrate tolerance range for the late-stage modification of natural

products and bioactive molecules in this reaction (Fig. 5). Encouragingly, a series of natural products (Leaf alcohol, β-Rhodinol, Geraniol, Fenchol, L(-)-Borneol, DL-Menthol, Diacetone-D-glucose) were all derivatized into the corresponding amino esters and installed into our substrates (**10a**–**10g**) which, upon treatment with ClCF$_2$COONa (**2d**)

under the optimized standard conditions, were all smoothly incorporated into the eventual Stevens rearrangement products (**11a**–**11g**), which showcased the viability of employing this protocol for a late-stage functionalization of complex amines.

## Synthetic utility

To demonstrate the synthetic utility of this difluorocarbene-induced Stevens rearrangement protocol, various post-functionalizations were employed (Fig. 6). The scalability was demonstrated by the successful gram-scale reaction of **3p**, **7d** and **9o**, and the corresponding products were generated without a significant decrease in yields, which further elucidates the practical value of the Stevens rearrangement reaction (Fig. 6a). Furthermore, the olefin products are highly versatile synthetic intermediates that can be readily transformed to an alkane in the presence of Pd/C and H$_2$[58], moreover, under the influence of Pd/C and H$_2$, the three-membered ring was simultaneously opened to yield a straight-chain propane **12**. The formamide product of this difluorocarbene-induced Stevens rearrangement can also be hydrolyzed with concentrated sulfuric acid[59] to obtain a secondary amine **13**. These scale-up experiments and the corresponding transformations of rearrangement products were successfully achieved, demonstrating the utility of the reaction and the diverse and useful transformations of the products.

## Mechanistic considerations

Based on previous literature[60,61] and experimental outcomes from this work, we speculate the possible mechanism of this difluorocarbene-induced tertiary amine-involved [1,2]- and [2,3]-Stevens rearrangement reactions (Fig. 7). The reaction first generates difluorocarbene in situ through the decomposition of difluoroalkylative reagent **2d**, which then combines with tertiary amine **A** to form the key active quaternary ammonium salt intermediate **B**. Direct completion of the [1,3]-proton shift via a strained four-membered ring transition state from **B** to **C** is very challenging based on the work proposed by Yu and colleagues in 2017[52,53], therefore, we think that protonation and deprotonation assisted by water are more feasible. The difluoromethyl carbanion on the quaternary ammonium salt **B** might be hydrolyzed in situ to lead to an amidyl species during this process under the action of base and water. Subsequent [2,3]- or [1,2]-sigmatropic rearrangement of **C** gives the corresponding rearrangement products through either concerted mechanism (for [2,3]-sigmatropic rearrangement) or radical pair path (for [1,2]-sigmatropic rearrangement) respectively[6–14].

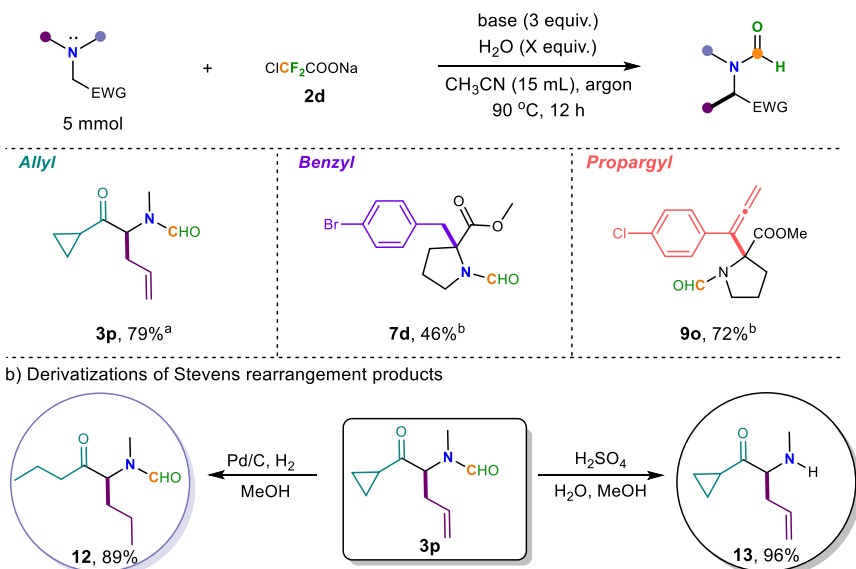

**Fig. 6 | Synthetic applications. a** Reactions performed at gram-scale.
**b** Derivatizations of Stevens rearrangement products. Reaction conditions: [a]**1p** (1 equiv., 5 mmol), **2d** (1.5 equiv., 7.5 mmol), H$_2$O (5 equiv.), K$_3$PO$_4$ (3 equiv.), CH$_3$CN (15 mL) at 90 °C for 18 h under argon. [b]**5d** or **8o** (1 equiv., 5 mmol), **2d** (1.5 equiv., 7.5 mmol), H$_2$O (15 equiv.), Rb$_2$CO$_3$ (3 equiv.), CH$_3$CN (2 mL) at 90 °C for 18 h under argon.

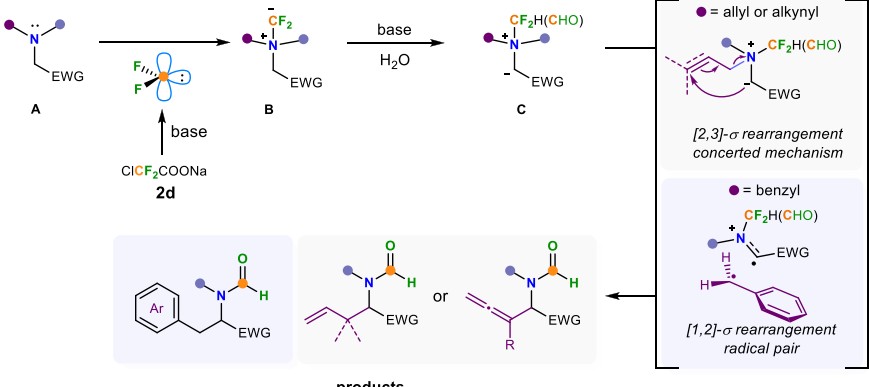

**Fig. 7 | Proposed mechanism.** Possible reaction mechanism of difluorocarbene-induced [1,2]- and [2,3]-Stevens rearrangement of tertiary amines.

In summary, we have reported a difluorocarbene-induced Stevens rearrangement reaction. Within this reaction system, various types of Stevens rearrangements could be accommodated, including the [2,3]-sigmatropic rearrangement of allylic and propargyl tertiary amines, as well as the [1,2]-sigmatropic rearrangement of benzylic tertiary amines. Additionally, a variety of migration products can be obtained efficiently. More importantly, the smooth conversion of this reaction further broadens the application of difluoromethyl ammonium salts, enabling Stevens rearrangement reactions beyond C-N bond activation, nucleophilic reagents, and deaminative cross-coupling. Given the simplicity and efficiency of these reactions, it is anticipated that this strategy would open up avenues to facilitate the exploration of such rearrangement reactions and contribute to the study of the properties of difluoromethyl ammonium salts.

## Methods

### Synthesis of 3

In air, tertiary amines **1** (1 eq, 0.2 mmol), $ClCF_2COONa$ (1.5 eq, 0.3 mmol), and $K_3PO_4$ (3 eq, 0.6 mmol) were added to a Schlenk tube equipped with a magnetic stirring bar. The vessel was evacuated and filled with argon (three cycles). $CH_3CN$ (2 mL) and $H_2O$ (5 eq, 1 mmol) were added by syringe under argon atmosphere. The resulting reaction mixture was stirred vigorously at 90 °C for 12 h. Upon completion of the reaction, the solvent was evaporated under reduced pressure, and the residue was purified by flash column chromatography (silica gel, petroleum ether: EtOAc = 2:1, v/v) to give the desired products.

### Synthesis of 6/7

In air, allylic amines **4** or benzyl amines **5** (1 eq, 0.2 mmol), $ClCF_2COONa$ (1.5 eq, 0.3 mmol), and $Rb_2CO_3$ (3 eq, 0.6 mmol) were added to a Schlenk tube equipped with a magnetic stirring bar. The vessel was evacuated and filled with argon (three cycles). $CH_3CN$ (2 mL) and $H_2O$ (15 eq, 3 mmol) were added by syringe under argon atmosphere. The resulting reaction mixture was stirred vigorously at 90 °C for 12 h. Upon completion of the reaction, the solvent was evaporated under reduced pressure, and the residue was purified by flash column chromatography (silica gel, petroleum ether: EtOAc = 2:1, v/v) to give the desired products.

### Synthesis of 9

In air, propargyl tertiary amines **8** (1 eq, 0.2 mmol), $ClCF_2COONa$ (1.5 eq, 0.3 mmol), and $Rb_2CO_3$ (3 eq, 0.6 mmol) were added to a Schlenk tube equipped with a magnetic stirring bar. The vessel was evacuated and filled with argon (three cycles). $CH_3CN$ (2 mL) and $H_2O$ (15 eq, 3 mmol) were added by syringe under argon atmosphere. The resulting reaction mixture was stirred vigorously at 90 °C for 12 h. Upon completion of the reaction, the solvent was evaporated under reduced pressure, and the residue was purified by flash column chromatography (silica gel, petroleum ether: EtOAc = 2:1, v/v) to give the desired products.

## Data availability

Data relating to the materials and methods, optimization studies, experimental procedures, NMR spectra, and mass spectrometry are available in the Supplementary Information. Crystallographic data for the structures reported in this Article have been deposited at the Cambridge Crystallographic Data Centre, under deposition number CCDC 2305294 (**9p**). Copies of the data can be obtained free of charge via https://www.ccdc.cam.ac.uk/structures/. Data can also be obtained from the corresponding author upon request.

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

## Acknowledgements

Financial support from the National Key R&D Program of China (2023YFF0723900), National Natural Science Foundation of China

(21931013 and 22271105), Natural Science Foundation of Fujian Province (2022J02009), and Open Research Fund of the School of Chemistry and Chemical Engineering, Henan Normal University is gratefully acknowledged.

## Author contributions

Q.S. conceived and directed the project. J.S. and Y.G. performed experiments and prepared the supplementary information. C.L. helped collecting some new compounds and analyzing the data. Q.S. and J.S. wrote the paper. All authors discussed the results and commented on the manuscript.

## Competing interests

The authors declare no competing interests.
