## [Peer Review File · Nature Communications]

Difluorocarbene-induced [1,2]- and [2,3]-Stevens rearrangement of tertiary aminesREVIEWER COMMENTS

Reviewer #1 (Remarks to the Author):

Please see the attached file for this reviewer's comment.

Reviewer #2 (Remarks to the Author):

In this manuscript, Song and coworkers reported the difluorocarbene-induced Stevens rearrangement for the synthesis of various tertiary amines. The reaction proceeds either via the [1,2] or [2,3] sigmatropic rearrangement pathway by the generation of the ammonium ylides. Typically, these reactions are performed under basic and harsh conditions, which limits the potential application of the method. The presented difluorocarbene route appears attractive and hence this referee recommends the publication of the work in Nature Communications after the revisions indicated below:

1. The scope of the reaction appears broad. However, it seems that the authors did not attempt the enantiospecific Stevens rearrangement. Such reactions starting from chiral proline derivatives should be tried and the results should be added.
2. The meaningful application of the product appears missing in the manuscript. The authors may apply this strategy for the easy access to a drug/natural product analog, which can considerably broaden the application of the method.
3. Similar to difluorocarbenes, arynes can induce similar Stevens rearrangements. Those papers should be cited in the manuscript: *Org. Lett.* 2016, 18, 5428-5431; *J. Org. Chem.* 2017, 82, 4470–4476.
4. Can the authors extend their strategy to benzyl/allyl thioethers for related Stevens rearrangement. This should be tried, and the results should be incorporated.
5. The authors may try to extend the reaction to related Sommelet-Hauser rearrangements as well.
6. Just like the difluorocarbenes, the reactive intermediates induce Stevens rearrangement is a known process, which can obviate the use of strong bases and harsh conditions. Related works should be presented in the introduction part of the manuscript.

Reviewer #3 (Remarks to the Author):

Song and co-workers have described herein a new protocol for the [1,2]- and [2,3]-Stevens rearrangements of tertiary amines. The [1,2]- and [2,3]-Stevens rearrangements constitute important methods for reorganizing nitrogen-containing compounds. Ammonium ylides are the key intermediates for [1,2]- and [2,3]-Stevens rearrangements, and they can be generated in situ by transition metal-catalyzed N-alkylation of tertiary amines with diazo compounds or allylic electrophiles. Another method for the in-situ generation of ammonium ylides is the reaction of tertiary amines with carbon electrophiles such as arynes, epoxides, and bicyclo[1.1.0]butanes. In this manuscript the authors employed in-situ generated difluorocarbene to activate tertiary amines to generate ammonium ylides, which allow the rearrangements of versatile tertiary amines bearing either allyl, benzyl, or propargyl groups. The reaction gave a range of rearrangement products in good yields. This work significantly extends the synthetic utilities of Stevens rearrangement reactions. In view of the significance of the work in the synthetic methodology, I recommend publication of this work in Nature Communications after revisions noted below.

- (1) A few closely relevant methods should be updated to the first paragraph on Page 2 for the in-situ generation of ammonium ylides as key intermediates for the [1,2]- and [2,3]-Stevens rearrangements. The ammonium ylides can also be generated in situ by the reaction of tertiary amines with carbon electrophile such as epoxides and bicyclo[1.1.0]butanes (see: Chem. Eur. J. 2019, 25, 5169-5172; Chem. Commun., 2021, 57, 8449-8451). Arynes have also been used to mediate the [2,3]-Stevens rearrangement of tertiary N-propargylamines (see: Chem. Commun., 2018, 54, 6036-6039).
- (2) At the bottom of Figure 1, the structure of the product is incorrect for the difluorocarbene-mediated Stevens rearrangement of a tertiary N-benzylamine. The phenyl group should be replaced with a benzyl group.
- (3) As shown in Figures 3 and 4, the Stevens rearrangement reaction gave products bearing a nitrogen-substituted quaternary stereocenter. Have the authors investigated the reaction of L-proline-derived tertiary amines? It would be interesting to achieve efficient central-to-central chirality transfer.
- (4) In Figure 2, EWG is an ester and a ketone. Can the reaction be extended to substrates bearing some other EWG such as an amide group and a cyano group?
- (5) In the Supporting Information, HRMS data should be included for the characterization of tertiary amines.

Song and colleagues have elucidated a difluorocarbene-induced [1,2]- and [2,3]-Stevens rearrangement reaction. As appropriately highlighted by the authors in both the abstract and introduction, the Stevens reaction has served as a prominent strategy for the molecular editing of common tertiary amines. Nevertheless, a set of universally applicable reaction conditions for the rearrangement of diverse moieties, including allyl, propargyl, and benzyl, has been conspicuously absent.

The Song group has been at the forefront of advancing the application of in-situ generated difluorocarbene in diverse transformations (ref 38). This paper details the innovative utilization of in-situ formed difluorocarbene intermediates for the synthesis of ammonium salt intermediates, subsequently leading to the generation of ammonium ylides in the presence of a base. The resulting ylides underwent Stevens rearrangement, showcasing versatility with various groups, including allyl, propargyl, and benzyl, in both [2,3]- and [1,2]-fashion, respectively. The outcomes presented herein not only contribute to rendering the Stevens rearrangement more user-friendly but also emphasize the extended utility of difluorocarbene chemistry. This research marks a significant step forward in the broader understanding and applicability of difluorocarbene-based methodologies.

However, this reviewer raises several concerns regarding the determination of whether this paper merits publication in prestigious and high-impact journals, such as Nature Communications. The authors aptly identified inherent challenges linked to utilizing difluorocarbene as a facilitator in the Stevens rearrangement reaction: 1) Steric hindrance posed by tertiary amines, 2) the potential emergence of a competing S_N2 pathway, and 3) the plausible hydrolysis of the difluoromethyl ammonium salt before the Stevens rearrangement occurs. Nevertheless, the reviewer contends that these challenges could be surmounted by employing relatively standard conditions for difluorocarbene generation, as previously utilized by the authors. In essence, the reviewer suggests that there is a lack of sufficiently "novel" aspects in the difluorocarbene chemistry contributing to the Stevens rearrangement. Instead, the established difluorocarbene chemistry seems to seamlessly apply to the Stevens rearrangement, portraying this as an incremental development rather than a groundbreaking discovery demanding immediate and significant attention from the synthetic community. Hence, this reviewer recommends the publication of this solid results to a journal that is more specialized for organic chemists.

The following typos should be fixed.

typo from Figure 1.

typo from Figure 7.

Point-by-point responses to the reviewers' comments

REVIEWER COMMENTS

Reviewer #1 (Remarks to the Author):

Song and colleagues have elucidated a difluorocarbene-induced [1,2]- and [2,3]-Stevens rearrangement reaction. As appropriately highlighted by the authors in both the abstract and introduction, the Stevens reaction has served as a prominent strategy for the molecular editing of common tertiary amines. Nevertheless, a set of universally applicable reaction conditions for the rearrangement of diverse moieties, including allyl, propargyl, and benzyl, has been conspicuously absent.

The Song group has been at the forefront of advancing the application of *in-situ* generated difluorocarbene in diverse transformations (ref 38). This paper details the innovative utilization of *in-situ* formed difluorocarbene intermediates for the synthesis of ammonium salt intermediates, subsequently leading to the generation of ammonium ylides in the presence of a base. The resulting ylides underwent Stevens rearrangement, showcasing versatility with various groups, including allyl, propargyl, and benzyl, in both [2,3]- and [1,2]-fashion, respectively. The outcomes presented herein not only contribute to rendering the Stevens rearrangement more user-friendly but also emphasize the extended utility of difluorocarbene chemistry. This research marks a significant step forward in the broader understanding and applicability of difluorocarbene-based methodologies.

Our Response: We sincerely thank this reviewer for his/her favorable comments on our submitted manuscript, we really appreciate it.

However, this reviewer raises several concerns regarding the determination of whether this paper merits publication in prestigious and high-impact journals, such as Nature Communications. The authors aptly identified inherent challenges linked to utilizing difluorocarbene as a facilitator in the Stevens rearrangement reaction: 1) Steric hindrance posed by tertiary amines, 2) the potential emergence of a competing S_N2 pathway, and 3) the plausible hydrolysis of the difluoromethyl ammonium salt before the Stevens rearrangement occurs. Nevertheless, the reviewer contends that these challenges could be surmounted by employing relatively standard conditions for difluorocarbene generation, as previously utilized by the authors. In essence, the reviewer suggests that there is a lack of sufficiently "novel" aspects in the difluorocarbene chemistry contributing to the Stevens rearrangement. Instead, the established difluorocarbene chemistry seems to seamlessly apply to the Stevens rearrangement, portraying this as an incremental development rather than a groundbreaking discovery demanding immediate and significant attention from the synthetic community. Hence, this reviewer recommends the publication of this solid results to a journal that is more specialized for organic chemists.

Our Response: We sincerely thank this reviewer for the critical yet inspiring comments on our submitted manuscript. As the reviewer pointed out, despite some developments in the Stevens rearrangement, notably the lack of universal conditions for various functionalities such as allylic, propargylic, and benzyl groups was evident. Building upon our previous research efforts, we continued to investigate the reactivity of the *in-situ* generated difluorocarbene quaternary ammonium salts from tertiary amines upon their reaction with difluorocarbene. Although we utilized a well-established system for generating difluorocarbene, we incorporated numerous design elements specific to this rearrangement reaction.

1. We conducted numerous trials on various rearrangement reactions facilitated by difluorocarbene, such as the Stevens rearrangement of thioethers and the Sommelet-Hauser rearrangement and others. Unfortunately, none of them yielded the desired products as expected. For thioether compounds, the substrates remained largely unreacted, while in the case of the Sommelet-Hauser rearrangement, direct S_N2 reactions occurred. After thorough screening of the rearrangement reactions, only the Stevens rearrangement of tertiary amines mediated by difluorocarbene proceeded smoothly.
2. This reaction carries the risk of directly generating allyl or benzyl halides. To prevent the direct S_N2 reactions and the formation of hydrolyzed products, we avoided using bromodifluoro reagents and iododifluoro reagents, which possess strong nucleophilicity. In this reaction, the difluorocarbene reagent must exhibit suitable reactivity to form the difluoromethyl ammonium ylide with the tertiary amine, meanwhile the departing halide ion should not be overly nucleophilic, as this would increase the likelihood of S_N2 reactions or direct hydrolysis to form amides, potentially terminating the reaction before the migration of the substituent. In this context, we hypothesized that chlorodifluoro reagents would be more suitable due to its less nucleophilicity. Therefore, we opted to employ chlorodifluoro reagent as the difluorocarbene source.

Through careful substrate selections, we successfully established a set of rearrangement conditions compatible with all allylic, propargylic, and benzyl functionalities, a feat which was not previously reported under universal conditions. This difluorocarbene-induced Stevens rearrangement not only enables skeletal editing of tertiary amines but also affords a diverse array of highly valuable compounds, including homoallylamines and α -substituted amine esters. Despite the considerable utility of these compounds, their syntheses have been relatively underexplored in previous reports. Leveraging this strategy allows for the mild conditions and strategic skeletal editing of tertiary amine compounds via the Stevens rearrangement, yielding homoallylamines and α -substituted amino esters. This provides a concise strategy for pharmaceutical applications, underscoring the practical value of this reaction.

This discovery and the transformations of tertiary amines are considered highly intriguing in synthetic chemistry, offering new avenues for research in rearrangement reactions. It opens up possibilities for exploring a wider range of reactive species and expands the scope of rearrangement reactions. Furthermore, it provides medicinal chemists with a concise and reliable strategy for synthesizing these frameworks, and

we believe it is suitable for *Nature Communications*. Here are some literature which demonstrate the applications of homoallylamines and α -substituted amino esters:

1. A new antitumor antibiotic, stubomycin. *J. Antibiot.* **1981**, *34*, 259.
2. Highly potent macrocyclic BACE-1 inhibitors incorporating a hydroxyethylamine core: design, synthesis and X-ray crystal structures of enzyme inhibitor complexes. *Bioorg. Med. Chem.* **2012**, *20*, 4377.
3. Recent advancements in the homoallylamine chemistry. *J. Heterocycl. Chem.* **2002**, *39*, 595.
4. Transition-metal-free allylation of 2-azaallyls with allyl ethers through polar and radical mechanisms. *Nat. Commun.*, **2021**, *12*, 3860.
5. A convenient synthesis of α -substituted cyclic α -imino acids. *Chem. Pharm. Bull.* **1979**, *27*, 1931.
6. Allosteric Wip1 phosphatase inhibition through flap-subdomain interaction. *Nat. Chem. Biol.* **2014**, *10*, 181.
7. Modular construction of unnatural α -tertiary amino acid derivatives by multicomponent radical cross-couplings. *Angew. Chem. Int. Ed.* **2022**, *61*, e202210755.

The following typos should be fixed.

typo from Figure 1.

typo from Figure 7.

Our Response: We sincerely thank this reviewer for carefully proofreading. Sorry for the negligence, we have made modifications. Please see our revised manuscript.

Reviewer #2 (Remarks to the Author):

In this manuscript, Song and coworkers reported the difluorocarbene-induced Stevens rearrangement for the synthesis of various tertiary amines. The reaction proceeds either via the [1,2] or [2,3] sigmatropic rearrangement pathway by the generation of the ammonium ylides. Typically, these reactions are performed under basic and harsh conditions, which limits the potential application of the method. The presented difluorocarbene route appears attractive and hence this referee recommends the publication of the work in *Nature Communications* after the revisions indicated below:

Our Response: We sincerely thank this reviewer for the favorable comments on our work. We really appreciate it.

1. The scope of the reaction appears broad. However, it seems that the authors did not attempt the enantiospecific Stevens rearrangement. Such reactions starting from chiral

proline derivatives should be tried and the results should be added.

Our Response: We deeply thank this reviewer for the questions. For the enantiospecific Stevens rearrangement, actually before the submission, we conducted some trials. However, the outcomes were not as anticipated. We synthesized various chiral tertiary amine substrates (including chiral proline and non-cyclic chiral amino acid esters), but only observed racemization of the chiral substrates during the reaction process. We speculated that this racemization might be induced by the base of the reaction. Subsequently, we attempted further optimization of the reaction conditions, but were unable to achieve satisfactory reaction outcomes.

And regarding this phenomenon, this time we also attempted to analyze the underlying reasons. Initially, we hypothesized that under such basic conditions, the α -hydrogen acidity of these amino acid esters might promote racemization. To investigate this, we conducted experiments under standard conditions without the addition of sodium chlorodifluoroacetate for both types of substrates. We observed complete racemization for non-cyclic amino acid esters, while no discernible effect was observed for cyclic amino acid esters.

Based on the above experiments, we have gained a deeper understanding of the mechanism of this reaction. 1) Although non-cyclic amino acid esters yield products with a relatively low enantiomeric excess after undergoing the rearrangement reaction (eq A), it still suggests the plausibility of [2,3]-sigmatropic rearrangement concerted mechanism. 2) Analysis of reactions B and D reveals that cyclic amino acid esters do

not racemize when treated with base alone, but under the standard conditions, they yielded completely racemized rearrangement products. This suggests that after the formation of intermediate difluoromethyl quaternary ammonium ylide C, the stronger acidity of the α -hydrogens makes racemization more easier and favorable, leading to this outcome. 3) At the same time, from the results of the above reactions, we can also infer that the rearrangement and racemization of intermediate C are competitive processes. For the substrates mentioned above, during the rearrangement of the alkene, the tertiary amine reacts with the difluorocarbene relatively quickly. At this point, the linear amino acid ester may not have racemized yet (or may not have fully racemized). When intermediate C is formed, the rearrangement occurs relatively quickly, resulting in the product being obtained with a relatively low degree of chirality retention. However, when the substrate is an alkyne, the rate of rearrangement is much slower than the rate of racemization, leading to complete racemization of the rearranged product.

Proposed mechanism.

$^1\text{H NMR}$ (500 MHz, Chloroform- d) δ 8.1 (d, J = 131.4 Hz, 1H), 7.7 – 7.6 (m, 2H), 7.2 – 7.0 (m, 2H), 5.7 (m, 1H), 5.2 – 5.0 (m, 4H), 3.0 (d, 3H), 2.9 (m, 1H), 2.6 – 2.5 (m, 1H), 1.5 (d, J = 67.7 Hz, 3H).

$^{13}\text{C NMR}$ (126 MHz, Chloroform- d) δ 172.8, 172.4, 162.8, 162.0, 137.9, 137.6, 135.6, 132.0, 130.5, 130.2, 130.2, 120.9, 119.9, 94.5, 93.9, 66.9, 66.3, 64.5, 61.7, 40.1, 39.6, 32.4, 28.5, 21.9, 20.5.

HPLC analysis: HPLC DAICEL CHIRALCEL OD-H, hexane/isopropanol = 95/5, 0.3 mL/min, λ = 210 nm, t_{R} (major) = 23.3 min, t_{R} (minor) = 31.7 min, er = 62:38.

Peak #	RetTime [min]	Type	Width [min]	Area [mAU*s]	Height [mAU]	Area %
1	23.152	BB	1.6883	6.12496e4	579.86633	49.3697
2	31.518	MM	3.5986	6.28137e4	290.91907	50.6303

Peak #	RetTime [min]	Type	Width [min]	Area [mAU*s]	Height [mAU]	Area %
1	23.257	BB	1.6768	4.27566e4	407.94855	61.6784
2	31.684	BB	2.6688	2.65652e4	127.92194	38.3216

¹H NMR (500 MHz, Chloroform-*d*) δ 8.3 – 8.1 (m, 1H), 5.5 (td, *J* = 6.7, 1.8 Hz, 1H), 5.0 – 4.9 (m, 2H), 3.7 (s, 3H), 3.7 – 3.5 (m, 2H), 2.5 (m, *J* = 12.9, 6.5, 2.0 Hz, 1H), 2.1 (m, 1H), 2.0 – 1.8 (m, 2H).

¹³C NMR (126 MHz, Chloroform-*d*) δ 207.3, 206.8, 172.3, 171.9, 162.6, 159.9, 93.8, 91.3, 80.2, 79.1, 67.7, 65.8, 53.1, 52.9, 47.2, 44.9, 38.5, 37.0, 23.4, 21.7.

HPLC analysis: HPLC DAICEL CHIRALCEL OD-H, hexane/isopropanol = 98/2, 0.3 mL/min, λ = 250 nm, *t*_R (major) = 27.9 min, *t*_R (minor) = 34.3 min, er = 50:50.

Peak #	RetTime [min]	Type	Width [min]	Area [mAU*s]	Height [mAU]	Area %
1	27.655	BB	0.8978	3.54505e4	601.32068	49.7186
2	34.101	BB	0.9948	3.58518e4	549.78992	50.2814

Peak #	RetTime [min]	Type	Width [min]	Area [mAU*s]	Height [mAU]	Area %
1	27.864	BB	0.8512	2473.67603	45.00821	50.0303
2	34.311	BB	0.9030	2470.68091	41.59611	49.9697

¹H NMR (500 MHz, Chloroform-*d*) δ 7.7 (d, $J = 8.3$ Hz, 2H), 7.1 (d, $J = 8.3$ Hz, 2H), 5.8 (dd, $J = 16.9, 10.3$ Hz, 1H), 5.2 – 5.0 (m, 4H), 3.5 (q, $J = 7.1$ Hz, 1H), 3.2 – 3.0 (m, 2H), 2.3 (s, 3H), 1.3 (d, $J = 7.1$ Hz, 3H).

¹³C NMR (126 MHz, Chloroform-*d*) δ 173.1, 137.7, 135.8, 135.7, 130.2, 117.7, 117.7, 94.0, 65.3, 60.5, 57.6, 37.7, 14.7.

HPLC analysis: HPLC DAICEL CHIRALCEL OJ-H, hexane/isopropanol = 99/1, 1 mL/min, $\lambda = 232$

nm, t_R (major) = 13.8 min, t_R (minor) = 15.3 min, er = 50:50.

1H NMR (500 MHz, Chloroform-*d*) δ 3.7 (s, 3H), 3.6 (dd, $J = 3.4, 2.3$ Hz, 2H), 3.4 (dd, $J = 9.2, 6.7$ Hz, 1H), 3.0 (ddd, $J = 9.5, 7.5, 2.6$ Hz, 1H), 2.7 (td, $J = 9.1, 7.5$ Hz, 1H), 2.2 (t, $J = 2.4$ Hz, 1H), 2.2 – 2.1 (m, 1H), 2.0 – 1.9 (m, 1H), 1.9 – 1.8 (m, 1H), 1.8 – 1.7 (m, 1H).

^{13}C NMR (126 MHz, Chloroform-*d*) δ 174.1, 73.2, 73.2, 62.5, 52.2, 52.0, 52.0, 41.2, 29.6, 23.3.

HPLC analysis: HPLC DAICEL CHIRALCEL OD-H, hexane/isopropanol = 99/1, 1 mL/min, $\lambda = 209$ nm, t_R (major) = 8.5 min, t_R (minor) = 9.4 min, er = 100:0.

2. The meaningful application of the product appears missing in the manuscript. The authors may apply this strategy for the easy access to a drug/natural product analog, which can considerably broaden the application of the method.

Our Response: We sincerely thank this reviewer for pointing it out and it is a great suggestion. We have conducted gram-scale reactions and several transformations to demonstrate the practical values of this reaction, enabling the conversion from tertiary amines to secondary amines. Analysis of the obtained products reveals a predominant formation of highly valuable classes of compounds, namely, homoallylamines¹⁻⁴ and α -substituted amino esters⁵⁻⁷, both are highly valuable compounds. Despite the considerable utility of these compounds, they have been relatively underexplored in previous synthetic methods. Leveraging this approach allows for the mild conditions and the strategic skeletal editing of tertiary amine compounds via the Stevens rearrangement, leading to the synthesis of homoallylamines and α -substituted amino esters. This offers a concise strategy for pharmaceutical applications, underscoring the practical values of this reaction. Here are some literature which demonstrate the applications of homoallylamines and α -substituted amino esters:

1. A new antitumor antibiotic, stubomycin. *J. Antibiot.* **1981**, *34*, 259.
2. Highly potent macrocyclic BACE-1 inhibitors incorporating a hydroxyethylamine core: design, synthesis and X-ray crystal structures of enzyme inhibitor complexes. *Bioorg. Med. Chem.* **2012**, *20*, 4377.
3. Recent advancements in the homoallylamine chemistry. *J. Heterocycl. Chem.* **2002**, *39*, 595.
4. Transition-metal-free allylation of 2-azaallyls with allyl ethers through polar and radical mechanisms. *Nat. Commun.*, **2021**, *12*, 3860.
5. A convenient synthesis of α -substituted cyclic α -imino acids. *Chem. Pharm. Bull.* **1979**, *27*, 1931.
6. Allosteric Wip1 phosphatase inhibition through flap-subdomain interaction. *Nat. Chem. Biol.* **2014**, *10*, 181.
7. Modular construction of unnatural α -tertiary amino acid derivatives by multicomponent radical cross-couplings. *Angew. Chem. Int. Ed.* **2022**, *61*, e202210755.

3. Similar to difluorocarbenes, arynes can induce similar Stevens rearrangements. Those papers should be cited in the manuscript: *Org. Lett.* 2016, *18*, 5428-5431; *J. Org. Chem.* 2017, *82*, 4470-4476.

Our Response: We deeply thank this reviewer for pointing it out, we have cited the article as ref. 19 (*J. Org. Chem.* **2017**, *82*, 4470-4476.) and ref. 21 (*Org. Lett.* **2016**, *18*, 5428-5431). Please see our revised manuscript.

3 Can the authors extend their strategy to benzyl/allyl thioethers for related Stevens rearrangement. This should be tried, and the results should be incorporated.

Our Response: We thank this reviewer for pointing it out, we prepared some thioether substrates (benzyl/allyl/alkyne thioethers), but found that these substrates were not compatible with the reaction. In all cases, no reaction occurred, and the substrates remained unchanged. The substrates used are shown in Part A of the following figure. In our previous substrate exploration, we observed that the presence of thioethers in the substrate structure inhibited the progress of the reaction as well, and no products were detected with all starting materials recovered. The substrates used are shown in Part B of the following figure.

4. The authors may try to extend the reaction to related Sommelet-Hauser rearrangements as well.

Our Response: We sincerely thank this reviewer for the constructive comments, it is a great suggestion. Per the request, we synthesized several suitable substrates based on literature references (*Chem. Commun.*, **2019**, *55*, 3004) and then subjected them to the reaction under the standard conditions. Unfortunately, we did not obtain the desired Sommelet-Hauser rearrangement products. Instead, the reactions yielded some residual starting materials and benzyl alcohols. We speculate that this product formation may stem from the activation of the tertiary amines, leading to benzyl chloride with its direct reaction with chloride ion present in the system. Subsequently, under the basic conditions, benzyl alcohol was formed. We also attempted reactions with electron-rich substituents and unsubstituted aromatic rings, but similar results were comparably encountered, failing to obtain the desired rearranged products. The examined substrates and the results are listed below:

5. Just like the difluorocarbenes, the reactive intermediates induce Stevens rearrangement is a known process, which can obviate the use of strong bases and harsh conditions. Related works should be presented in the introduction part of the manuscript.

Our Response: We thank this reviewer for pointing it out. We have incorporated the relevant content into the article. Please see our revised manuscript.

Reviewer #3 (Remarks to the Author):

Song and co-workers have described herein a new protocol for the [1,2]- and [2,3]-Stevens rearrangements of tertiary amines. The [1,2]- and [2,3]-Stevens rearrangements constitute important methods for reorganizing nitrogen-containing compounds. Ammonium ylides are the key intermediates for [1,2]- and [2,3]-Stevens rearrangements, and they can be generated in situ by transition metal-catalyzed N-alkylation of tertiary amines with diazo compounds or allylic electrophiles. Another method for the in-situ generation of ammonium ylides is the reaction of tertiary amines with carbon electrophiles such as arynes, epoxides, and bicyclo[1.1.0]butanes. In this manuscript the authors employed in-situ generated difluorocarbene to activate tertiary amines to generate ammonium ylides, which allow the rearrangements of versatile tertiary amines bearing either allyl, benzyl, or propargyl groups. The reaction gave a range of rearrangement products in good yields. This work significantly extends the synthetic utilities of Stevens rearrangement reactions. In view of the significance of the work in the synthetic methodology, I recommend publication of this work in Nature Communications after revisions noted below.

Our Response: We sincerely thank this reviewer for his/her favorable comments on our manuscript, we really appreciate it!

(1) A few closely relevant methods should be updated to the first paragraph on Page 2 for the in-situ generation of ammonium ylides as key intermediates for the [1,2]- and [2,3]-Stevens rearrangements. The ammonium ylides can also be generated in situ by the reaction of tertiary amines with carbon electrophile such as epoxides and bicyclo[1.1.0]butanes (see: *Chem. Eur. J.* 2019, 25, 5169-5172; *Chem. Commun.*, 2021, 57, 8449-8451). Arynes have also been used to mediate the [2,3]-Stevens rearrangement of tertiary N-propargylamines (see: *Chem. Commun.*, 2018, 54, 6036-6039).

Our Response: We deeply thank this reviewer for pointing it out, we have cited the article as ref. 27 (*Chem. Eur. J.* 2019, 25, 5169-5172.), ref. 26 (*Chem. Commun.*, 2021, 57, 8449-8451.) and ref. 22 (*Chem. Commun.*, 2018, 54, 6036-6039.). Please see our revised manuscript.

(2) At the bottom of Figure 1, the structure of the product is incorrect for the difluorocarbene-mediated Stevens rearrangement of a tertiary N-benzylamine. The phenyl group should be replaced with a benzyl group.

Our Response: We sincerely thank this reviewer for pointing it out, sorry for the negligence, we have corrected this issue. Please see our revised manuscript.

(3) As shown in Figures 3 and 4, the Stevens rearrangement reaction gave products bearing a nitrogen-substituted quaternary stereocenter. Have the authors investigated the reaction of L-proline-derived tertiary amines? It would be interesting to achieve efficient central-to-central chirality transfer.

Our Response: We deeply thank this reviewer for the questions. For the enantiospecific Stevens rearrangement, actually before the submission, we conducted some trials.

However, the outcomes were not as anticipated. We synthesized various chiral tertiary amine substrates (including chiral proline and non-cyclic chiral amino acid esters), but only observed racemization of the chiral substrates during the reaction process. We speculated that this racemization might be induced by the base of the reaction. Subsequently, we attempted further optimization of the reaction conditions, but were unable to achieve satisfactory reaction outcomes.

And regarding this phenomenon, this time we also attempted to analyze the underlying reasons. Initially, we hypothesized that under such base conditions, the α -hydrogen acidity of these amino acid esters might promote racemization. To investigate this, we conducted experiments under standard conditions without the addition of sodium chlorodifluoroacetate for both types of substrates. We observed complete racemization for non-cyclic amino acid esters, while no discernible effect was observed for cyclic amino acid esters.

Based on the above experiments, we have gained a deeper understanding of the mechanism of this reaction. 1) Although non-cyclic amino acid esters yield products with a relatively low enantiomeric excess after undergoing the rearrangement reaction (eq A), it still suggests the plausibility of [2,3]-sigmatropic rearrangement concerted mechanism. 2) Analysis of reactions B and D reveals that cyclic amino acid esters do not racemize when treated with base alone, but under the standard conditions, they yielded completely racemized rearrangement products. This suggests that after the formation of intermediate difluoromethyl quaternary ammonium ylide C, the stronger

acidity of the α -hydrogens makes racemization more easier and favorable, leading to this outcome. 3) At the same time, from the results of the above reactions, we can also infer that the rearrangement and racemization of intermediate C are competitive processes. For the substrates mentioned above, during the rearrangement of the alkene, the tertiary amine reacts with the difluorocarbene relatively quickly. At this point, the linear amino acid ester may not have racemized yet (or may not have fully racemized). When intermediate C is formed, the rearrangement occurs relatively quickly, resulting in the product being obtained with a relatively low degree of chirality retention. However, when the substrate is an alkyne, the rate of rearrangement is much slower than the rate of racemization, leading to complete racemization of the rearranged product.

Proposed mechanism.

$^1\text{H NMR}$ (500 MHz, Chloroform-*d*) δ 8.1 (d, $J = 131.4$ Hz, 1H), 7.7 – 7.6 (m, 2H), 7.2 – 7.0 (m, 2H), 5.7 (m, 1H), 5.2 – 5.0 (m, 4H), 3.0 (d, 3H), 2.9 (m, 1H), 2.6 – 2.5 (m, 1H), 1.5 (d, $J = 67.7$ Hz, 3H).

$^{13}\text{C NMR}$ (126 MHz, Chloroform-*d*) δ 172.8, 172.4, 162.8, 162.0, 137.9, 137.6, 135.6, 132.0, 130.5, 130.2, 130.2, 120.9, 119.9, 94.5, 93.9, 66.9, 66.3, 64.5, 61.7, 40.1, 39.6, 32.4, 28.5, 21.9, 20.5.

HPLC analysis: HPLC DAICEL CHIRALCEL OD-H, hexane/isopropanol = 95/5, 0.3 mL/min, $\lambda = 210$ nm, t_{R} (major) = 23.3 min, t_{R} (minor) = 31.7 min, er = 62:38.

Peak #	RetTime [min]	Type	Width [min]	Area [mAU*s]	Height [mAU]	Area %
1	23.152	BB	1.6883	6.12496e4	579.86633	49.3697
2	31.518	MM	3.5986	6.28137e4	290.91907	50.6303

Peak #	RetTime [min]	Type	Width [min]	Area [mAU*s]	Height [mAU]	Area %
1	23.257	BB	1.6768	4.27566e4	407.94855	61.6784
2	31.684	BB	2.6688	2.65652e4	127.92194	38.3216

$^1\text{H NMR}$ (500 MHz, Chloroform-*d*) δ 8.3 – 8.1 (m, 1H), 5.5 (td, $J = 6.7, 1.8$ Hz, 1H), 5.0 – 4.9 (m, 2H), 3.7 (s, 3H), 3.7 – 3.5 (m, 2H), 2.5 (m, $J = 12.9, 6.5, 2.0$ Hz, 1H), 2.1 (m, 1H), 2.0 – 1.8 (m, 2H).

$^{13}\text{C NMR}$ (126 MHz, Chloroform-*d*) δ 207.3, 206.8, 172.3, 171.9, 162.6, 159.9, 93.8, 91.3, 80.2, 79.1, 67.7, 65.8, 53.1, 52.9, 47.2, 44.9, 38.5, 37.0, 23.4, 21.7.

HPLC analysis: HPLC DAICEL CHIRALCEL OD-H, hexane/isopropanol = 98/2, 0.3 mL/min, $\lambda = 250$ nm, t_{R} (major) = 27.9 min, t_{R} (minor) = 34.3 min, er = 50:50.

Peak #	RetTime [min]	Type	Width [min]	Area [mAU*s]	Height [mAU]	Area %
1	27.655	BB	0.8978	3.54505e4	601.32068	49.7186
2	34.101	BB	0.9948	3.58518e4	549.78992	50.2814

Peak #	RetTime [min]	Type	Width [min]	Area [mAU*s]	Height [mAU]	Area %
1	27.864	BB	0.8512	2473.67603	45.00821	50.0303
2	34.311	BB	0.9030	2470.68091	41.59611	49.9697

¹H NMR (500 MHz, Chloroform-*d*) δ 7.7 (d, $J = 8.3$ Hz, 2H), 7.1 (d, $J = 8.3$ Hz, 2H), 5.8 (dd, $J = 16.9$, 10.3 Hz, 1H), 5.2 – 5.0 (m, 4H), 3.5 (q, $J = 7.1$ Hz, 1H), 3.2 – 3.0 (m, 2H), 2.3 (s, 3H), 1.3 (d, $J = 7.1$ Hz, 3H).

¹³C NMR (126 MHz, Chloroform-*d*) δ 173.1, 137.7, 135.8, 135.7, 130.2, 117.7, 117.7, 94.0, 65.3, 60.5, 57.6, 37.7, 14.7.

HPLC analysis: HPLC DAICEL CHIRALCEL OJ-H, hexane/isopropanol = 99/1, 1 mL/min, $\lambda = 232$

nm, t_R (major) = 13.8 min, t_R (minor) = 15.3 min, er = 50:50.

1H NMR (500 MHz, Chloroform-*d*) δ 3.7 (s, 3H), 3.6 (dd, $J = 3.4, 2.3$ Hz, 2H), 3.4 (dd, $J = 9.2, 6.7$ Hz, 1H), 3.0 (ddd, $J = 9.5, 7.5, 2.6$ Hz, 1H), 2.7 (td, $J = 9.1, 7.5$ Hz, 1H), 2.2 (t, $J = 2.4$ Hz, 1H), 2.2 – 2.1 (m, 1H), 2.0 – 1.9 (m, 1H), 1.9 – 1.8 (m, 1H), 1.8 – 1.7 (m, 1H).

^{13}C NMR (126 MHz, Chloroform-*d*) δ 174.1, 73.2, 73.2, 62.5, 52.2, 52.0, 52.0, 41.2, 29.6, 23.3.

HPLC analysis: HPLC DAICEL CHIRALCEL OD-H, hexane/isopropanol = 99/1, 1 mL/min, $\lambda = 209$ nm, t_R (major) = 8.5 min, t_R (minor) = 9.4 min, er = 100:0.

(4) In Figure 2, EWG is an ester and a ketone. Can the reaction be extended to substrates bearing some other EWG such as an amide group and a cyano group?

Our Response: We deeply thank this reviewer for pointing it out and it is a great suggestion. In terms of other electron-withdrawing groups (not an ester or a ketone), when an amide was employed as the electron-withdrawing group, it failed to induce the Stevens rearrangement reaction. Instead, we observed the formation of a product where the tertiary amine undergoes direct allylic cleavage at the nitrogen atom upon interaction with difluorocarbene species, followed by formylation.

Fortunately, cyano group can serve as suitable electron-withdrawing groups to promote the rearrangement reaction as well. We also explored whether phosphate esters could serve as suitable electron-withdrawing groups and delightedly found that they can successfully yield the desired product with allylic migration. We have incorporated these results into our new main text. Please see our revised manuscript.

(5) In the Supporting Information, HRMS data should be included for the characterization of tertiary amines.

Our Response: We apologize for the negligence. We have added HRMS data of tertiary amine in the SI. Please see our revised manuscript and supporting information.

REVIEWERS' COMMENTS

Reviewer #2 (Remarks to the Author):

The authors submitted the revised version of the manuscript for further consideration. All the comments of this referee and the other referees have been satisfactorily taken care of by the authors and sufficient modifications have been carried out during revision. Publication in the present form is recommended. Congratulations!

Reviewer #3 (Remarks to the Author):

I have reviewed this manuscript before. The authors have addressed the reviewers' comments and queries. I recommend publication of the current version in Nature Communications.

Point-by-point responses to the reviewers' comments

REVIEWER COMMENTS

Reviewer #2 (Remarks to the Author):

The authors submitted the revised version of the manuscript for further consideration. All the comments of this referee and the other referees have been satisfactorily taken care of by the authors and sufficient modifications have been carried out during revision. Publication in the present form is recommended. Congratulations!

Our Response: We sincerely thank this reviewer for his/her favorable comments on our manuscript, we really appreciate it!

Reviewer #3 (Remarks to the Author):

I have reviewed this manuscript before. The authors have addressed the reviewers' comments and queries. I recommend publication of the current version in Nature Communications.

Our Response: We sincerely thank this reviewer for the favorable comments on our work. We really appreciate it.